# Reality Check: A New Evaluation Ecosystem Is Necessary to Understand AI's Real World Effects

## Abstract

Conventional AI evaluation approaches concentrated within the AI stack exhibit systemic limitations for exploring, navigating and resolving the human and societal factors that play out in real world deployment such as in education, finance, healthcare, and employment sectors. AI capability evaluations can capture detail about first-order effects, such as whether immediate system outputs are accurate, or contain toxic, biased or stereotypical content, but AI's second-order effects, i.e. any long-term outcomes and consequences that may result from AI use in the real world, have become a significant area of interest as the technology becomes embedded in our daily lives. These secondary effects can include shifts in user behavior, societal, cultural and economic ramifications, workforce transformations, and long-term downstream impacts that may result from a broad and growing set of risks. **This position paper argues that measuring the indirect and secondary effects of AI will require expansion beyond static, single-turn approaches conducted in silico to include testing paradigms that can capture what actually materializes when people use AI technology in context.** Specifically, we describe the need for data and methods that can facilitate contextual awareness and enable downstream interpretation and decision making about AI's secondary effects, and recommend requirements for a new ecosystem.

## 1 Introduction

As AI technologies have become mainstream, the number of tools for evaluating them have exploded within a highly active and competitive area of development and research. Measurement provides AI practitioners with the opportunity to test and learn whether and how the technology they build works once deployed [1] Evaluation enables interpretation of measurement results to place them into context. Metrology, the science of measurement, provides the methods and definitions of measurement that enable the evaluation of all measurement results, including for AI systems. Metrology provides the foundations for estimating measurement uncertainty that can incorporate multiple sources of random and systematic error.

AI testing and evaluation is currently conducted within a computational and machine learning (ML) frame, with few systematic methods to account for the complex human, organizational and societal factors that inter-relate with the design, development, deployment and use of these technologies. This socio-technical [2] framing of AI technology is currently difficult for ML practitioners to operationalize,

---

[1]Measurement: (1) Quantitative measurement is the act or process of assigning a number or category to an entity to describe an attribute of that entity. ISO/IEC 24765:2017 (2) Qualitative measurement is based on descriptive data such as through observations, interviews, focus groups, or open-ended text fields in surveys.

[2]The term "socio-technical systems" was coined in 1951 by Eric Trist and Ken Bamforth[101]to describe the dynamic ways workers interact with technological systems in industrial settings.

Table 1: Mapping evaluation approaches to effects measured and typical questions they answer.

| Evaluation Approach | Type of Effects (order) | What it measures | Answers questions like |
|---|---|---|---|
| Benchmarking | 1st | Performance of the model/system *in silico*. | 1. How often can the AI system produce the most accurate or relevant answer?
2. What is the inference runtime?
3. Did the model produce human-aligned responses? |
| Testing & Evaluation | 1st, 2nd | Performance of the model/system *in silico in vitro* and *in situ*. | 1. Does text summarization provide value for users?
2. Given current performance and user needs, should we expect productivity gains if we deploy this technology? If so, where? |
| Verification & Validation | 1st, 2nd | Performance of the model/system *in silico*, *in vitro* and *in situ.*. | 1. Does the AI system consistently generate video content per user specifications?
2. Does the AI system classify output according to vendor claims? |
| Program Evaluation | 2nd, 3rd | Real-world efficacy and relevance *in vitro* and *in situ*. | 1. Do AI assistants improve the quality of work?
2. How will AI-driven productivity gains transform different employment categories over time? |

or to know where, when and how to include which types of contextual information across the technology lifecycle. **This paper argues that a new AI evaluation ecosystem is necessary to address current methodological gaps which impede the translation and contextualization of evaluation data and outcomes in the real world [107, 104, 16, 27, 29, 33, 85, 83, 35, 96, 37, 81, 76].** A real world AI evaluation ecosystem can enhance understanding of AI's second-order effects, drive the collection of datasets that are fit-for-purpose, foster innovation, and improve AI functionality.

## 1.1 The Measurement Challenge

The speed at which AI technology is advancing and being deployed and used across the globe [102] is not being met with equivalent evaluation paradigms for understanding its role and effects in societies. As a central topic of public policy efforts around the globe, questions about AI's secondary effects abound. Private industry, civil society, the public, and governments around the world are increasingly interested in how AI technologies will transform our culture, economy, workforce and the broader society[75, 74, 72, 73, 40, 43]. The current ecosystem to investigate these topics is fragmented, with no single evaluation toolbox or measurement infrastructure to account for AI's second order effects and place them into the broader context.

The predominant evaluation toolbox used by the ML research community, AI benchmarking, is designed to answer *first order* questions– about *what* AI systems can do based on direct measures of immediate system output. Another broad set of domains study AI's human, organizational and societal factors, which tend to focus more on second order questions such as the effects associated with how people leverage AI technology, and how and why those effects reverberate across society. Other fields can place these findings into context to forecast future technological and societal trends. Some approaches, like user simulations, can simultaneously model AI user behavior and evaluate system performance. The AI metrology community is also deeply engaged in the development of tools to assess systems in more realistic settings with the broader goal of ensuring AI system trustworthiness. This work includes development of definitions [10], and methods for calibration [106], and uncertainty quantification [39, 103] and propagation [100]. Yet, more effort is required, including efficient scalability and interpretation of measurement values.

Table 1 lists differences between the kinds of questions that can be answered by benchmarking, testing and evaluation, verification and validation, and program evaluation respectively. As methods

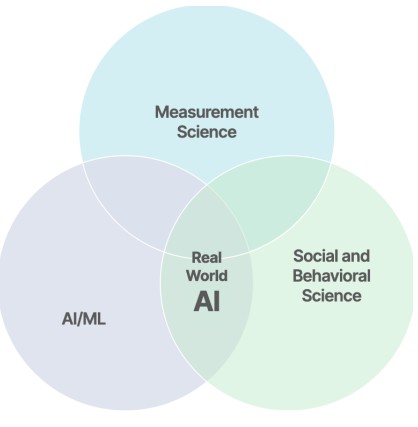

,

Figure 1: Disciplines at the intersection of Real World AI

move from first to second order and contextual detail increases, broader claims about AI technology become possible [1].

All evaluation methods have limitations which, when combined with the massive heterogeneity of how humans interact with AI in the real world, can present almost infinite complexities [71, 33] for building comprehensive paradigms. In addition to the technical and human factors of what and how to evaluate, there are numerous disciplinary and practical challenges to contend with. Reproducibility is an AI evaluation challenge of particular interest in computational domains and the social and behavioral sciences. In machine learning , reproducibility challenges include data leakage issues [51], non-systematic methods for curating training data, non-disclosure of training data information, unstable model versioning processes [50], and insufficient detail about experimental design, metadata, and related analytic processes[30, 80]. Advances in the culture of research practice have emerged to address the replicability crisis in the social and behavioral sciences [17]. Experiment pre-registration, open science standards, multiverse and sensitivity analyses, meta-analyses, and adversarial collaborations have led to varying levels of improvement [54, 6].

While a new ecosystem does not eliminate the above-listed challenges, a purpose-built community can concentrate on improving methods for evaluating second order effects. Figure 1 illustrates a real world AI community at the intersection of AI and ML, measurement science, and the social and behavioral sciences which can adapt and re-purpose methods, tools, metrics and practices to fuel deeper understanding of AI's complex societal challenges. This interdisciplinary community can collaboratively establish relevant measurement criteria, collect suitable datasets, formalize methods and practices and use resulting insights to produce better models for automation and real world forecasting and decision making.

## 2 AI Benchmarking

Model benchmarking is the de facto AI evaluation method. Benchmarking uses static datasets to assess performance of AI model capabilities on specific tasks at scale, often in comparison to humans. Evaluators use benchmark results to compare different models or systems on the same tasks. Benchmark suites are used to aggregate results and comprehensively assess capabilities, risks, and compliance. For example, systems may be tested for truthfulness [109], toxicity [42], and jailbreak vulnerability [24]. Benchmarking outcomes underpin AI system design, procurement, and oversight activities.

---

[1]In the context of AI evaluation, 1st-order effects are immediate system outputs, 2nd-order effects are longer-term impacts that may follow from system deployment, 3rd-order effects refer to broader changes that may result from AI's role in society. *In silico* refers to testing conducted via computational methods. *In situ* refers to observing a phenomenon in its natural location or context. *In vitro* refers to traditional laboratory experiments

## 2.1 The Practice of Benchmarking

The current benchmark landscape spans a wide range of tasks (text generation, question answering, summarization), modalities (text, code, audio, images, video), and evaluation dimensions such as factuality, fairness, safety, and alignment with human preferences[93, 14, 105, 109, 87, 63, 23, 53]. Recent benchmarks are built on various risk taxonomies to support an increased focus on AI risks and safety. For example, safety benchmarks can explicitly target risks posed by prompt injection[24] and data leakage [32], or be used to assess subtle failure modes that require multiple tests to capture nuanced system characteristics[82].

Benchmarks serve multiple purposes in the current evaluation landscape. They inform the setting of minimum performance thresholds-often through system requirements, regulatory norms, or technical standards but rarely define what constitutes "adequate" performance for a given policy context. With benchmarking defining the very notion of "success", these tests can shape perceptions of AI progress, influence research and development priorities, and inform investment cycles[35]. For this to be meaningful, benchmarking must be consistent and conducted before, during and after the development and deployment of AI tools and systems.

Benchmark design is complex but typically starts with identifying or acquiring curated datasets and specifying controlled tasks. Most generative AI benchmarks are conducted at the 'single-turn' level and assess independent query/response pairs[61]. Multi-turn benchmarks can be used to simulate more realistic dialogue and provide richer insights. Benchmarks rely on highly structured outputs such as multiple-choice, or short paragraph responses). Since there are technical limitations to which measures and characteristics can be analyzed at scale, evaluators often use LLMs to judge LLM benchmark output[110].

Benchmark test results are typically displayed via leaderboards, which provide a structured way to rank model performance and ease comparison [105, 89]. Undesirably, the evaluation community's overreliance on leaderboards can lead to overfitting (models are optimized for test performance at the expense of real-world robustness) or to benchmark saturation, where further improvements on the test no longer translate to meaningful advances[99].

## 2.2 Selected Benchmarking Limitations for Real World Evaluation

Benchmarking's prominence in AI evaluation has propelled the community to groundbreaking improvements and fostered global innovation but the outcomes can be limited. Benchmarking requires significant contextualization to serve the decision making needs of the many audiences interested in AI's second order effects. AI benchmarks have been criticized for lacking internal and external validity [77, 64], encouraging leaderboard overfitting[78], and focusing narrowly on English-language tasks [70]. More broadly, they suffer from static design, lack of systematization, limited stakeholder involvement, and a failure to reflect cultural nuance. The static nature of benchmarking may obscure emergent behaviors, security vulnerabilities, or context-specific failures that only surface in deployment or over longer time periods. It is difficult to construct benchmarking tasks that naturally elicit generative AI risks – such as harmful bias, hallucinations, or user over-reliance – [83, 1, 86, 13, 49, 31, 25] and their associated impacts[77, 7], or the broad range of user responses and behavior that may arise from LLM-based personalization. These limitations can lead to skewed perceptions of AI's real-world use and value[52].

Even within an active community, benchmarks are unable to capture the full array of AI system functionality and performance. Benchmarking can lag behind new model capabilities, especially for complex agentic tasks or qualitative aspects like creativity and reasoning. Benchmarks can be prone to task contamination or data leakage resulting in erroneously high performance [51, 60]. The intense interest in generative AI model capabilities has driven the use of tests that were designed for other purposes (e.g., testing models on college admission tests or professional certifications) – or poorly specify human tasks, potentially distorting perceptions of progress[47, 67, 3]. Benchmarks are designed to mathematically represent complex human and societal phenomena, which can contribute to a fallacy of objectivity [83]. Benchmarking metrics focus on system accuracy or policy violations, which are challenging to apply to second order questions.

Arguably the biggest weakness of benchmarking is its inability to account for the inter-dependencies between humans and AI, such as how people leverage AI or interpret and act upon AI-generated output in the real world, and what it means at a societal level. Even the most comprehensive benchmark suites remain abstractions that offer only partial glimpses into real world effects[71]. The need for

scale has led to reliance on static benchmark datasets and highly constrained tasks which are a poor match for deployment environments where contextual factors and user perceptions can dramatically alter outcomes [107, 27, 33, 85].

# 3 Context is Everything: Crafting a Real World AI Evaluation Ecosystem

The ability to make claims about the real world requires authentic and extensive contextual detail. Contextual awareness - knowledge about what matters in a given deployment setting - can improve AI's fit within societal contexts and foster measurement validity. Contextual information can fulfill two requirements for real world AI evaluation that benchmarking struggles to address. First, non-ML actors use this information to translate and make sense of evaluation results for their own activities and decision making. Second, practitioners on the AI stack can gain complementary evidence of how the technology they build is actually being used in deployment.

Currently, sensing and leveraging contextual information from the real world is impeded by processes in the AI stack. While ML models can be derived from trillions of data points, the development process flattens contextual detail. Recent approaches to align model outcomes to predefined and prescriptive values [5] reduce societal and contextual detail instead of eliciting and analyzing it. Many organizations also lack the skills and methods to interpret and translate contextual material from the real world (such as user reviews, information from redress and recourse, other stakeholder feedback) into AI product workflows[90]. Combined, these practices can bake in brittle performance once AI systems are deployed [83, 16, 55, 21, 56, 66].

This section describes methods for how to specify context for real world evaluation and to collect and generate contextually-informed data. Methods for analyzing contextual information will be the focus of future directions.

**Establishing Contextual Awareness**
The field of value-sensitive design (VSD) and its tripartite methodology (conceptual, empirical, technical) provides a foundational framework to operationalize contextual awareness for real world AI evaluation. Table 2 summarizes how practices for specifying contextual scope and collecting real world data fit into the VSD framework.

| Contextual Approach | Key Integration Practices | Outcome | VSD Method |
|---|---|---|---|
| **Context Specification** | Initiate Theory of Change Systematize Real World Concepts Stakeholder Engagement | Contextually informed requirements for data collection and generation activities. | Conceptual |
| **Data Collection and Generation** | Field Testing Red Teaming | Data about regular and adversarial use of AI systems. | Empirical Technical |

Table 2: Overview of context-aware AI evaluation approaches and their interdisciplinary roles.

By docking into the VSD framework, real world AI evaluation methods can produce continuous feedback loops–where context specification activities inform red teaming (to identify real-world failures) and field testing (to determine extent of failures in regular use). Since red teaming and field testing enable investigation of "the technology, the people who use it, and the social systems that configure, use, or are otherwise affected by the technology"[38] it satisfies both the empirical and technical VSD methods. VSD processes can also assist in translating evaluation outcomes into technical/policy adjustments.

## 3.1 Context Specification Activities

The activities described below define the real world challenge problem, the context in which it exists, and other relevant detail. Gathering this information is the first step in facilitating contextual awareness and requires input from a broad set of stakeholders to ensure measurement validity.

### 3.1.1 Theory of Change

Real world AI evaluation activities are initiated by defining a theory of change. Key stakeholders and evaluators collaboratively identify challenge problems, specify desired goals over the current state and determine evaluation inputs, activities, outputs, and outcomes. Stakeholders also assist evaluators in identifying counterfactuals to estimate what might happen without the evaluation effort.

### 3.1.2 Systematization of Real World Concepts

Real world concepts that underlie the development of an AI model's objective function and other variables drive system functionality, optimization and performance. The validity of an AI model can hinge on how well these real world concepts are systematized and operationalized [26], which requires technical, neutral, collectively informed and unambiguous descriptions. Models that do not demonstrate validity cannot maintain performance well across contexts. Systematized descriptions can be used to:

- instruct AI models to properly recognize a given phenomenon and act accordingly in context,
- optimize development of prompts for user engagement with AI systems and to ensure model outcome meets preferences and requirements,
- enhance content markup and moderation for complex and ambiguous phenomena (e.g., obscenity, abusive or hateful content).

Currently, ML practitioners demonstrate difficulty with systematization and operationalization, and it is challenging to bridge the communication divide between computational and other disciplines and translate real world concepts along product lifecycles [36, 91, 34, 66, 92] .

### 3.1.3 Stakeholder Feedback and Adaptive Governance

AI evaluators are increasingly exploring methods that better reflect deployment conditions and integrate members of the public directly into the measurement process. Meaningful stakeholder engagement methods are a common component of adaptive AI governance frameworks [59, 28, 11] and can bolster public accountability, democratic governance, and transparency efforts such as recourse and redress. Engagement is conducted throughout the entire AI project lifecycle and can effectively inform evaluation activities. Engagement activities use a variety of qualitative methods to capture a range of perspectives and experiences from stakeholders external to the AI development organization. Stakeholder engagement activities can be built into evaluation paradigms to facilitate contextual awareness [57, 8, 68, 48] by:

- revealing potential negative impacts prior to AI development and deployment and shed light on unanticipated AI uses and positive outcomes,
- surfacing emergent risks or gradual declines in real world system performance
- informing mitigation of AI harms before they become entrenched, [62, 85, 4]
- surfacing assumptions and limitations about AI technology.

The Alan Turing Institute's AI Sustainability in Practice workbook [59] lays out a stakeholder engagement process which begins with a determination of the groups most likely to be negatively impacted by AI systems. The level of subsequent stakeholder involvement–ranging from inform or consult to partner or empower–is proportionate to the scope of a project's potential risks and impacts [59]. Participatory co-creation is another engagement method that moves beyond traditional consultation to enable and empower stakeholders in more active roles across the AI design, development, deployment processes. Stakeholders work closely with AI designers from the initial context specification phase, iterate on the design and user interface, support the creation of governance structures, and inform system monitoring [8].

## 3.2 Collecting and Generating Contextually-Informed Data

Once the contextual unit of interest has been defined, data collection activities can be designed and executed. Two methods for collecting and generating contextually informed data – field testing and red teaming– are described below. While benchmarking relies almost entirely on curated and labeled

datasets, red teaming and field testing can be used to design and collect response data from different types of audiences as they interact with AI systems in the real world.

### 3.2.1 Field Testing

Field methods and experiments have been used by social scientists for decades to gain insights into human and social behavior by bridging laboratory settings and the real world. Methods similar to field testing[1] are regularly used in technology settings but its adapted use in AI evaluation is relatively nascent, with recent work in the field of AI risk assessment [84, 79]. Designed to elicit and capture detailed information about what happens under regular use, field testing is conducted through empirical observation of individuals as they interact with AI technologies under semi-controlled conditions across multiple sessions. While the focus of benchmarking is the AI model or system, AI field testing can focus on the "contextual unit" – or the complex and adaptive behavior that naturally occurs as people leverage AI technology in setting. Field testing can be used to explore how humans use and adapt to AI technology, investigate feedback loops between humans and technology, [27, 98, 41], and uncover emergent or "long-tail" scenarios that single-turn, lab-based benchmarks might miss.

In a simulated sandbox and reporting environment[2], hundreds or even thousands of human subjects interact live with AI systems and provide feedback about their experiences and subsequent actions. Resulting dialogues from test interactions can be annotated to determine whether various phenomena materialized. This descriptive reporting approach transforms evaluation paradigms beyond whether or not a system generated "the right answer" or asking people to judge AI output or train AI systems. Instead, field testing enables the collection of real world evidence about what materializes when certain AI features are deployed to the broader public. Since field tests are conducted in a controlled and protected environment, evaluators can safely configure pre-deployment testing suites and responsibly explore a wide variety of factors. When using field testing to measure accuracy of system responses, task contamination and data leakage are less likely than in benchmarking due to the difficulty of anticipating the heterogeneous prompts of thousands of testers.

Field testing requires:

- Multi-session experiments to observe how subjects adapt to AI technology over repeated usage (days or weeks).
- Experimental randomization and blinding to minimize biases in user interactions or system responses.
- Observation and analysis of subject responses and behaviors alongside isolated system outputs such as user surveys, logs, and performance metrics.
- Test scenarios for subject interactions with AI systems that balance naturalistic conditions and subject safety [79].
- Human subject research protocols.
- Descriptive approaches for marking up interactive output [79].

### 3.2.2 Red Teaming

The rise in generative AI use and its associated impacts has contributed to increased interest in AI red teaming as a complement to conventional evaluation paradigms[108]. Unlike static benchmarks, red teaming can simulate real-world usage to

- uncover failures, trends and patterns that emerge in complex or adversarial settings,
- highlight misuse and weaknesses in system behavior and robustness,
- determine boundary conditions to inform go/no-go decisions about deploying AI, and
- verify the effectiveness of existing mitigation strategies, safety measures and frameworks.

Red teaming is often conducted via "challenges", where individual testers use simulated attacks to identify vulnerabilities and evaluate the safety and security of AI systems. Red teamers may use

---

[1]such as A/B tests

[2]Can also be referred to as a large-scale human testbed

creative multi-turn prompting, role-playing, and other techniques to probe the model's responses and surface undesirable model outputs, such as data leakage [1], jailbreaking [2], and information based harms[3] Red teaming challenges can surface detailed information about how harmful outcomes occur, who they affect, how they circulate in social contexts or are repurposed by malicious actors, and how system vulnerabilities evolve over time [98, 22]. Red teaming is especially valuable in high-stakes domains like education, healthcare, and employment, where harms may be severe or emerge gradually, or disproportionately impact marginalized groups.

Red teaming challenges require detailed instructions, rules of engagement, and a framework, policy, or set of rules for identifying violative outcomes. Various tasks along the AI pipeline may require individuals to engage with harmful test scenarios or to be exposed to toxic and violent content, and red teaming is no different. To protect red teamer safety, challenges require appropriate psychological safety mechanisms to be put in place prior to participant enrollment.

Red teaming requires diverse backgrounds and domain expertise to cover the broad range of potential harms posed by AI systems. For example, multi-lingual expertise is required to test AI systems for linguistic and dialectal biases and gaps in language coverage. Challenges can go beyond simple Q&A tasks to test models on summarization and translation tasks and sentiment analysis. Red Teaming challenges may entail:

- **Expert Red Teaming:** Highly skilled professionals with expertise in adversarial misuse or exploits, or in the underlying subject matter, simulate sophisticated attacks to identify deep-seated vulnerabilities.
- **Public Red Teaming:** Members of the general public interact with AI systems under controlled or "challenge" conditions to complement expert red teaming and expand the tested risk surface. Public participants do not require expertise in adversarial testing but instead seek to surface real-world failures or "off-label" uses that expert red teamers may not anticipate or consider, such as how AI systems may fail across cultural or linguistic contexts.
- **Automated Red Teaming:** The automated generation of adversarial prompts or test cases at scale to uncover issues such as data leakage or content policy circumvention. Evaluators can automate parts of the red teaming process to expand test coverage and reveal systemic model weaknesses.

Challenge designers can combine public and expert-based red teaming exercises into hybrid challenges and leverage principles of collective intelligence, where testers can coordinate with – or learn from – each other's discoveries.[4] Collaborative and asynchronous exercises can encourage knowledge-sharing and expedite the discovery of edge cases. Manual and automated techniques can also be combined to balance the strengths and limitations of both approaches[69]. Red teaming can be used alongside field testing to determine whether adversarial vulnerabilities may manifest in regular use, or if new ones arise from repeated user queries

**Red Teaming Attack Strategies** In addition to the list of red teaming attack strategies found in Appendix A, red teamers can systematically employ data poisoning, indirect prompt injection, or multi-turn "scenario chaining" to force AI systems into unforeseeable states and capture vulnerabilities that may only appear after multiple interactions or under disguised prompts. Periodic red teaming "rounds" can be used to track whether system updates inadvertently open up new exploits or degrade previously solved safeguards.

**Selected Red Teaming Limitations** As AI systems evolve, red teaming efforts can adapt through interdisciplinary development of new attack vectors and multi-turn or multi-modal tasks. [46, 98]. A list of recommendations that challenge designers can use to address selected red teaming limitations is provided below:

---

[1]Revealing sensitive information from AI system training data.

[2]Circumventing safety measures and generating restricted, privileged, dangerous, copyrighted and/or otherwise unauthorized material.

[3]Obscene, degrading, abusive, and radicalizing material; content that may not distinguish fact from fiction; content that may amplify, reify or exacerbate biases against different sub-groups or lead to disparities between sub-groups; false content that may mislead or deceive users (aka hallucinations).

[4]Small groups of experts can collectively overcome a learning curve faster than individuals, allowing them to identify more subtle or complex vulnerabilities in a shorter time frame.[22, 95]

- **Scoping** Address scoping limitations by including multi-turn conversations, multiple languages and dialects, and multi-modal tasks.

- **Tester Biases** Address participant bias and representation issues by expanding the red teaming recruitment process beyond traditional settings, broadening dataset requirements, surveying red teamer perceptions of harm, and introducing positionality statements.

- **Automation** Collaboratively develop criteria for automated generation of high-quality, diverse test cases while preserving the nuanced understanding of human red teamers.

- **Resource Constraints** Balance the cost and efficiency of manual red teaming with scalable but limited automated approaches to ensure engagement from smaller organizations or research groups.

- **Transparency and Information Sharing** Establish guidelines for the responsible, open and transparent sharing of red teaming findings that take ethical implications and potential misuse into account.

- **Evaluating Effectiveness** Build off of information security red teaming metrics[1] to collaboratively define criteria for desirable and undesirable system behavior and advance evaluation metrics and methods to track progress over time.

## 4  Summary and Recommendations

Policy makers, organizational decision makers and members of the public each require different types of information about AI so they can make informed decisions about whether and how to develop, deploy or use it in their own contexts. A real world AI evaluation ecosystem to support these audiences will have to contend with many trade-offs to gather information beyond the AI stack and within context.

While benchmarking is too limited on its own to investigate second order effects, other types of evaluation that provide more fidelity are disconnected from the necessary system measurements central to AI benchmarking. "Contextual work" is commonly viewed as slow and resource-intensive compared to benchmarking, since it requires different processes, actors, skills and disciplines. For example, fielding qualitative research surveys and conducting ethnographies are both more expensive and time-consuming than using "found data". Activities surrounding problem specification are also consistently overlooked due to a perception that they take too long and don't provide enough benefit.

Both red teaming and field testing require infrastructure that can host people and technology in deployed scenarios while meeting human subject research requirements. All evaluation methods will require transparency, reproducibility, and scientific integrity. Even when built on feedback from thousands of people, evaluation outcomes do not automatically ladder up to societal insights such as impacts to democracy, the workforce and the economy, education, and culture[45, 97, 15, 2, 65].

With no existing infrastructure or community dedicated to evaluating AI's second order effects, other procedural models could be used as exemplars. A new ecosystem could be supported through the creation of testing hubs that include expertise from academia, industry, and civil society to develop rigorous science-backed evaluation methodologies and frameworks. Ecosystem inputs could be sourced from organizations that bring their questions to bear. Members of the public could support specification of contextual inputs and enroll in red teaming and field testing activities. Organizations that have relevant evaluation expertise and methods can provide their services as independent testers to enhance credibility and ensure objectivity in the evaluation process. The academic research community can support the development of formalized metrics and methodologies. Over time, the ecosystem can determine which evaluation activities produce value and should be automated (and semi-automated) to enhance scalability and adoption.

Outputs from ecosystem activities will center on answering second order effects and fostering a more dynamic and adaptive real world AI evaluation community. Anticipated insights will include deeper understanding of how AI technologies function outside tightly controlled lab settings, how users might abuse or misunderstand AI functionality and outputs, and how AI's role in society influences systemic trends.

---

[1]such as incident detection rate, time to detect incident, and mean time to recovery

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

# Appendix

## A  Selected Red Teaming Attack Strategies

- Complex or leading prompts can expose common AI system vulnerabilities such as confabulation, logically inconsistent responses, faulty reasoning, flawed decision-making, incorrect numeric responses, erroneous code generation, and fabricated citations.
- Counterfactual prompting and the use of repeated requests while varying demographic personas can uncover harmful biases [9].
- Autocompletion, fill-in-the-blank requests and prompts designed as "honest" requests can be used to evaluate system guardrails and force AI systems to produce harmful completions.
- Membership inference attacks, and probes of training data memorization can be used to expose sensitive or private information [18–20, 32, 94].
- Prompting for sensitive personal or location-based details can be used to evaluate data handling and privacy safeguards.
- Combining jailbreaking attacks with counterfactual prompts in multiple languages and dialects can be used to force culturally and linguistically biased output.
- Data poisoning, indirect prompt injection, misleading training inputs [58], and embedding harmful prompts subtly within benign content [44] can be used to evaluate system integrity and resistance to manipulation.
- Availability or "sponge" attacks use excessively large numbers of queries to stress test AI systems for performance stability and resource resilience [88].
- Chaos testing and random attacks expose systems to excessively large numbers of random prompts to elicit failures or jailbreaks (these prompts can be AI generated).
- Adversarial examples and membership inference attacks are used to probe security vulnerabilities [18–20, 32, 94].
- Prompts for copyrighted or proprietary content can be used to surface intellectual property risks [12].
- Prompts for obscene or abusive content can be used to evaluate the efficacy of content moderation.

