# OpenReview forum: "Reality Check: A New Evaluation Ecosystem Is Necessary to Understand AI's Real World Effects"
_NeurIPS.cc/2025/Position_Paper_Track — Submitted to NeurIPS 2025 Position Paper Track_

### Official Review · Reviewer_gC8A · 2025-07-10

**Significance:** 4
**Presentation:** 3
**Rating:** 7
**Confidence:** 4

**Summary:**

This position paper argues that current AI evaluation methods, primarily benchmarking, are inadequate for understanding AI's real-world societal impacts. The authors contend that while benchmarking captures "first-order effects" (immediate system outputs), it fails to measure "second-order effects" (long-term consequences of AI deployment). They propose a new interdisciplinary evaluation ecosystem incorporating field testing, red teaming, and value-sensitive design principles. The paper calls for moving beyond static, computational evaluation toward dynamic, contextually-aware methods that involve stakeholders and capture how humans actually interact with AI systems in deployment. The authors recommend establishing testing hubs with expertise from academia, industry, and civil society to develop more comprehensive evaluation frameworks that can inform policy and deployment decisions.

**Strengths:**

1. The paper tackles a genuinely important problem with AI evaluation that has real implications for AI safety and governance. The interdisciplinary approach is refreshing and necessary for this complex topic.

2. The systematic critique of benchmarking limitations is thorough and well-reasoned. The proposed framework, integrating value-sensitive design with field testing and red teaming, is innovative and practically oriented.

3. The writing demonstrates broad expertise across multiple relevant fields. The paper successfully makes the case that current evaluation approaches are insufficient for understanding AI's societal impacts, which is a crucial insight for the field.

4. The extensive literature review shows deep engagement with relevant work across disciplines.

**Weaknesses:**

1. The paper would benefit from more concrete examples of successful implementations of the proposed methods. While the authors critique benchmarking limitations, they could better address potential counterarguments about the scalability and cost-effectiveness of their proposed alternatives.

2. The paper lacks a discussion of how to standardize or validate the proposed contextual evaluation methods. Some claims about benchmarking limitations may be overstated - benchmarking has evolved significantly, and some recent work does attempt to address real-world relevance.

3. The paper could better acknowledge the practical challenges of implementing their proposed ecosystem, including resource requirements and institutional barriers. The transition between describing problems and proposing solutions could be smoother.

**Questions:**

1. How would you envision standardizing field testing and red teaming methodologies across different organizations and domains to ensure comparable and reliable results?

2. What specific mechanisms would you propose to balance the need for comprehensive contextual evaluation with the practical constraints of time and resources that organizations face?

3. Could you provide more details on how the proposed evaluation ecosystem would handle the challenge of keeping pace with rapidly evolving AI capabilities while maintaining methodological rigor?

**Alternative Position:**

Yes, and alternative positions are well-considered and named but not addressed

**Author Identification:**

No.

**Context:**

4

**Details Of Ethics Concerns:**

The paper extensively discusses field testing and red teaming methodologies that involve human participants interacting with AI systems. While the authors acknowledge the need for "human subject research protocols" and "psychological safety mechanisms," the paper lacks sufficient detail about how these studies would comply with IRB requirements. The proposed "hundreds or even thousands of human subjects" participating in field tests raises concerns about informed consent, participant safety (especially when exposed to potentially harmful AI outputs), and data privacy. The red teaming challenges described could expose participants to "toxic and violent content," requiring more robust ethical safeguards than currently outlined.

**Discussion:**

4

**Ethics:**

["Major Concern: Improper research involving human subjects"]

**Position:**

Yes, the paper argues for or against a position related to machine learning.

**Support:**

3

**Thoroughness:**

5

---

### Official Review · Reviewer_H6Ja · 2025-08-02

**Significance:** 3
**Presentation:** 3
**Rating:** 5
**Confidence:** 4

**Summary:**

The authors contend that current AI evaluation methods, particularly benchmarking, fall short in capturing contextual information, referred to as second-order effects. They further argue that, when these limited evaluation methods are combined with the vast heterogeneity in how humans interact with AI in real-world settings, the result is a near-infinite complexity that hampers the development of a comprehensive evaluation framework. Reproducibility is also identified as a major challenge, as existing benchmarks tend to be static in design, lack systematic structure, involve limited stakeholder input, and fail to account for cultural nuance. Therefore, the authors advocate for an AI evaluation ecosystem that is contextually aware and grounded in real-world dynamics.

**Strengths:**

The authors clearly state their position: the current evaluation system, which relies heavily on AI benchmarks, fails to capture the social dynamics between AI and humans. They argue that context awareness should be grounded in value-sensitive design (VSD) principles and supported by key integration practices such as stakeholder engagement, field testing, and red teaming, among others.

**Weaknesses:**

The authors did not substantiate their proposed solutions with concrete, contextual examples from real-world applications, such as text generation and object detection etc. While the third-order effect is briefly mentioned in Table 1, the paper lacks any meaningful discussion of its relevance to AI evaluation systems. Additionally, the authors overlook a significant body of ongoing research that addresses the societal impact of AI algorithms, as well as issues of reproducibility and explainability, leaving important perspectives unacknowledged.

**Questions:**

1. Does the proposed AI ecosystem aim to replace current benchmarking methods, such as leaderboards, or complement them?

2. How can human bias be mitigated when human input directly influences the development and evaluation of AI algorithms?

3. Could building an AI evaluation system grounded in contextual factors—such as culture or discipline—hinder progress toward developing artificial general intelligence?

4. If AI evaluation is based on contextual information, how can a standardized, uniform evaluation framework be established across diverse applications?

**Alternative Position:**

Yes, and alternative positions are well-considered and addressed by the argument

**Author Identification:**

No.

**Context:**

3

**Discussion:**

3

**Ethics:**

["NO or VERY MINOR ethics concerns only"]

**Position:**

Yes, the paper argues for or against a position related to machine learning.

**Support:**

2

**Thoroughness:**

4

---

### Official Review · Reviewer_JASC · 2025-08-07

**Significance:** 3
**Presentation:** 3
**Rating:** 4
**Confidence:** 4

**Summary:**

Existing AI evaluation benchmarks primarily focus on first-order effects (e.g., does the AI system produce accurate predictions?), while second- and third-order effects (e.g., long-term downstream impacts on users and society) remain largely unexplored. The paper argues that a new ecosystem should be developed to measure these indirect impacts of AI systems.

**Strengths:**

1. Effectively highlights the limitations of current AI evaluation practices (e.g., model benchmarking on static datasets).

2. Calls for a new context-aware AI measurement and impact evaluation ecosystem involving stakeholders from both AI and non-AI domains.

**Weaknesses:**

1. Since the limitations of current AI benchmarking practices are highlighted, it would be helpful to discuss specific technical solutions that the NeurIPS community should consider.

2. Human-Computer Interaction (HCI) and Human-AI Interaction (HAI) researchers already actively study the impact of AI on user behavior. They incorporate tactics mentioned in the position paper, such as red teaming and conduct field experiments. The authors should clarify what specifically is missing from current HCI/HAI research.

3. The proposed measurement/evaluation ecosystem lacks specific details and remains high-level. The paper would benefit from providing concrete examples of how contextual data adds value in specific domains (such as medical decision-making with AI versus AI support in education).

**Questions:**

1. How can the NeurIPS community contribute to developing a more effective and useful measurement/evaluation system?

2. What gaps exist in current HCI/HAI research regarding the measurement of AI's impact on users and society? What specific, actionable steps could address these limitations?

3. What are some specific examples of leveraging contextual data in certain domains? Could the authors provide a clearer picture with examples?

**Alternative Position:**

Yes, and alternative positions are trivial straw-man arguments

**Author Identification:**

No.

**Context:**

3

**Discussion:**

3

**Ethics:**

["NO or VERY MINOR ethics concerns only"]

**Position:**

Yes, the paper argues for or against a position related to machine learning.

**Support:**

3

**Thoroughness:**

3

---

### Note · Authors · 2025-09-04

**1-11 Submit Again:**

Probably yes

**1-1 Submission Process:**

4

**1-4 Interest:**

["Panel discussions with other position paper authors", "Mentorship programs for early-career researchers"]

**1-5 Thoughtful:**

7

**1-6 Supportive:**

6

**1-7 Technical Aspects Versus Position:**

6

**1-8 Gate Keeping:**

8

**1-9 Camera Ready Changes:**

If accepted, we will clarify that the paper’s primary contribution is proposing a new lens on AI evaluation that foregrounds second- and third-order effects rather than offering a detailed technical blueprint. We will add short domain vignettes (e.g., education, healthcare) to show how contextual evaluation changes outcomes, and expand our discussion of third-order societal impacts. We will also strengthen the paper by addressing standardization and feasibility, outlining how shared reporting formats can ensure comparability while keeping methods practical and adaptive. We will also provide a brief summary on how HCI efforts differ from the approaches described in the paper.

**3-1 Review Response1:**

JASC

**3-2 Reaction To Review1:**

We thank the reviewer for the thoughtful feedback. This paper aims to advance a new lens on AI evaluation that foregrounds second and third-order effects, rather than provide a full technical blueprint.

We acknowledge the contributions and role of HCI and HAI in efforts such as red teaming and field experiments. Yet, these approaches emphasize individual or group behavior without incorporating the measurement science practices (eg: uncertainty quantification, traceability, and reproducibility) that we argue are essential for evaluating societal impacts beyond localized user or system outputs.

Our proposed ecosystem builds on HCI/HAI by combining their methods with formal measurement principles. Contextual data should come not only from user studies but also from organizational, cultural, and societal settings to enable more robust evaluation. Human-AI dialogues, for instance, offer linguistic signals that complement HCI’s behavioral focus. In healthcare, such data might show how AI alters workflows and outcomes; in education, it could capture longer-term effects on curriculum and equity beyond short-term satisfaction metrics.

Despite strengths in capturing user experience, HCI/HAI often lacks mechanisms to scale individual-level insights to societal outcomes, account for long-term feedback loops, and apply measurement rigor to support reproducibility and policy relevance.

The NeurIPS community can help address these gaps by integrating HCI/HAI methods with measurement science, supporting domain-specific evaluation hubs, reusable contextual data, and collaborative testbeds that leverage AI developers, domain experts, and stakeholders to co-design real-world challenge problems.

Space constraints limited our ability to elaborate on use cases and recommendations. We focused on benchmarking due to its prominence in current evaluation practice. We hope this paper sparks broader dialogue and welcome collaboration on applied, domain-specific evaluation in future work.

**3-3 Review Response2:**

H6Ja

**3-4 Reaction To Review2:**

We thank the reviewer for the thoughtful comments and for the opportunity to expand on our position.

On benchmarking. Our proposed ecosystem complements rather than replaces benchmarking. Benchmarks capture first-order system performance but must be paired with contextual methods to guide downstream decision-making. We’ll highlight this relationship in the camera-ready version.

On human bias. We agree that human processes across the lifecycle, within organizations, and end-user perceptions can cause harm. Rather than abstracting these factors, we argue they should be surfaced and managed. A socio-technical frame, supported by metrology-inspired practices like uncertainty quantification and stakeholder diversity, can help systematize this.

On AGI and contextual grounding. We view contextual evaluation as an AGI enabler.  AGI must operate across varied environments, yet current systems struggle with real-world diversity. Contextual awareness acts as a sensor—systematically capturing the diverse data necessary to support modeling and analysis. Our approach leverages interdisciplinary expertise to foster open exchange of assumptions, methods, and perspectives, enhancing robustness and innovation. Integrating such insights helps ensure AGI is both capable and responsibly deployed.

On standardization. Contextual evaluation strengthens, not weakens, comparability. Frameworks can be general yet adaptable to specific settings. Without structured ways to incorporate context, practitioners must reinvent methods for each domain. As with benchmarks, we envision formats (e.g., “context cards”) documenting deployment factors—enabling comparability while surfacing nuance.

On examples and third-order effects. We agree domain-specific examples would add value. Due to space limits, this paper is a starting point. Follow-up work will explore applied use cases. We hope this sparks further discussion and collaboration within the interdisciplinary AI evaluation community.

**3-5 Review Response3:**

gC8A

**3-6 Reaction To Review3:**

We appreciate the reviewer’s thoughtful reading and strong endorsement of the paper’s importance.
On ethics and human subjects. We agree that red teaming and field testing raise ethical considerations and will reaffirm that safeguards are a prerequisite—not an afterthought—and reference existing guidance in our response.
On standardization. We envision standardization of field testing and red teaming via shared processes, evaluation challenges, and common reporting formats (e.g., context cards, uncertainty estimates, safety protocols). Standardization need not mean uniformity—rather, it should enable comparability and learning. Community-wide challenges can drive understanding of the practices that yield better outcomes. As in benchmarking, teams would put their methods to the test on shared real-world tasks and be scored on performance. Results would identify the best combinatorial evaluation approaches by domain. NeurIPS could help support such collaborative testing at scale.

On balancing comprehensiveness and feasibility. We don’t propose replacing benchmarks, but augmenting them with contextual methods. The depth of contextualization can scale with stakes and resources—from lightweight stakeholder feedback to longitudinal fieldwork.
On AI’s rapid pace. Our proposed ecosystem is adaptive: modular standards, reusable protocols, and shared testbeds make updating evaluations easier as models evolve—no need to reinvent methodology each time.

We will clarify these points in the revision, and briefly address cost, scalability, and implementation constraints for a more balanced argument.

---

### Meta-Review · Area_Chair_JyVH · 2025-09-12

**Rating:** 5
**Confidence:** 3

**Strengths:**

The paper tackles a significant and timely problem by arguing that current AI benchmarking fails to capture the critical second- and third-order societal impacts of AI systems. The call for a new, interdisciplinary, and context-aware evaluation ecosystem is compelling and well-reasoned, drawing effectively on principles like value-sensitive design. The paper demonstrates a broad understanding of the limitations of current methods and the need to incorporate stakeholder engagement, field testing, and red teaming.

**Weaknesses:**

The weakness is a lack of concrete detail and actionable steps. The proposed ecosystem remains high-level and abstract, lacking specific examples of how it would work in practice for different AI domains (e.g., medical vs. educational AI). The paper also underacknowledges extensive existing work in HCI and ethics that already studies these impacts, failing to clearly articulate what gap their specific proposal fills beyond this existing research. A concern was also raised regarding the insufficient detail on safeguarding the "hundreds or even thousands" of human participants proposed for field tests and red teaming.

**Questions:**

A central tension in the proposal is between context-specific evaluation and the need for standardized, comparable results. How to balance these two purposes in evaluation effectively and efficiently?

**Ethics:**

Major Concern: Improper research involving human subjects

**Thoroughness:**

3

---

### Decision · Program_Chairs · 2025-09-26

Reject